# Comparative Analysis of Tetra(2-naphthyl)tetracyano-porphyrazine and Its Iron Complex as Photosensitizers for Anticancer Photodynamic Therapy

**DOI:** 10.3390/pharmaceutics14122655

**Published:** 2022-11-30

**Authors:** Lydia N. Shestakova, Tatyana S. Lyubova, Svetlana A. Lermontova, Artem O. Belotelov, Nina N. Peskova, Larisa G. Klapshina, Irina V. Balalaeva, Natalia Y. Shilyagina

**Affiliations:** 1Institute of Biology and Biomedicine, Lobachevsky State University, Gagarin Ave., 23, 603950 Nizhny Novgorod, Russia; 2Razuvaev Institute of Organomettalic Chemistry, Russian Academy of Sciences, St. Tropinina, 49, 603137 Nizhny Novgorod, Russia

**Keywords:** photodynamic therapy, PDT, photosensitizer, cancer treatment, porphyrazine, Fe(II) complex, cyanoarylporphyrazines

## Abstract

Photodynamic therapy (PDT) is a rapidly developing modality of primary and adjuvant anticancer treatment. The main trends today are the search for new effective photodynamic agents and the creation of targeted delivery systems with the function of controlling the release of the agent in the tumor. Recently, the new group of cyanoarylporphyrazine dyes was reported, which combine the properties of photosensitizers and sensors of the local microenvironment. Such unique characteristics allow the release of the photosensitizer from the transport carrier to be assessed in real time in vivo. The aim of the present work was to compare the photophysical and photobiological properties of tetra(2-naphthyl)tetracyanoporphyrazine and its newly synthesized Fe(II) complex. We have shown that the chelation of the Fe(II) cation with the porphyrazine macrocycle leads to a decrease in molar extinction and an increase in the quantum yield of fluorescence and photostability. We demonstrate that the iron cation significantly affects the rate of dye accumulation in cells, the dark toxicity and photodynamic activity, and the direction of the changes depends on the particular cell line. However, in all the cases, the photodynamic index of a metal complex was higher than that of a metal-free base. In general, both of the compounds were found to be very promising for PDT, including for the use with transport delivery systems, and can be recommended for further in vivo studies.

## 1. Introduction

Photodynamic therapy (PDT) was developed over forty years ago and is now widely used to treat superficial tumors or tumors accessible with an endoscope. PDT is used as an independent treatment method as well as an adjuvant therapy modality along with traditional methods of tumor treatment, such as chemotherapy, radiation therapy and surgery [1]. This therapeutic procedure involves the administration of a photosensitive compound (photosensitizer, PS), which accumulates in the tumor. Then, tumor cells are exposed to visible or near infrared (NIR) light of the wavelength that corresponds to the peaks of the PS absorption spectrum [2]. The excited PS reacts with oxygen, which leads to the generation of reactive oxygen species (ROS), such as singlet oxygen, superoxide anion, peroxide, hydroxyl radical, hydroperoxide radical, etc. The generated ROS trigger oxidative reactions, which lead to the death of tumor cells [3]. The anticancer effect of PDT is based on three main mechanisms: the direct photodynamic effect on tumor cells, damage to the blood vessels that feed the tumor, and a systemic immune response [4,5].

At one of the first stages in the development of PDT, the criteria for an ideal PS were formulated. An ideal photosensitizer should have the following properties: low dark and high light toxicity, high selectivity of accumulation in target cells and rapid elimination from healthy tissues, high quantum yield of production of reactive oxygen species that cause the death of the target cells [6].

The main efforts in the field of experimental PDT have been focused on the screening of potential PS with different chemical structures. In vitro and in vivo studies demonstrate that the physico-chemical properties of PS significantly affect the processes of PS localization and its photodynamic activity in cells and tumor tissue [7]. Tetrapyrroles, such as hematoporphyrins [8], porphyrins [9,10,11], chlorins [12,13,14] and bacteriochlorins [15], purpurins [16,17], porphyrazines (tetraazaporphyrins) [18,19,20] and others have been studied as the main potential photosensitizers (PSs).

One of the approaches to the modification of a photosensitizer is an approach based on the inclusion of various metals in the center or to peripheral framing of the macrocycle. The presence and nature of the central metal ion strongly influences the photochemical, photophysical, and photobiological properties of the PS [21], such as water solubility, triplet state lifetime, intersystem crossing efficiency, quantum yield and fluorescence lifetime, and photostability [22,23].

The most common metals used in the modification of tetrapyrrolic PSs are aluminum (Al), iron (Fe), zinc (Zn), copper (Cu), tin (Sn), manganese (Mn), and nickel (Ni) [23]; the less common metals are heavy metals such as ruthenium (Ru), iridium (Ir), platinum (Pt), lutetium (Lu), palladium (Pd), Rhodium (Rh), thulium (Tm), and ytterbium (Yb) [24,25].

Although metal complexes of photosensitizers have been studied for a long time, only two of them have passed clinical trials. They are TOOKAD^®^ soluble (Padeliporfin, WST11), which is a palladium-coordinated bacteriochlorophyll a, approved for the treatment of prostate cancer in Mexico, Israel and over 30 EU countries [26]; and Photosens^®^, which is sulfonated aluminum phthalocyanine approved for the treatment of skin, liver, breast, lung and gastrointestinal cancer in Russia [27]. Several photosensitizer metal complexes are currently going through clinical trials: Purlytin^®^ (Rostaporfin), a tin-coordinated phthalocyanine PS, which is in clinical trials in the USA for the treatment of basal cell carcinoma [28]; Antrin^®^ (Motexafin Lutetium), a texaphyrin-based compound with a lutetium metal center and axially bound acetate ions for the treatment of prostate cancer, brain cancer, breast cancer, and cervical cancer [29,30]; and ruthenium polypyridine complex TLD-1433, clinically tested in Canada for invasive bladder cancer [31].

Another approach to increasing the effectiveness of PDT is the development of delivery systems for photosensitizers [32]. Nanoscale carriers are of particular interest as delivery systems. They can provide an enhancement in PS solubility and prolonged blood circulation of PS, and facilitate PS targeted delivery and controlled release in the pathological site [33,34,35,36,37]. Targeted accumulation of photosensitizers in the tumor helps to reduce the serious side effects of PDT, such as prolonged (over several months) photosensitivity of the skin and eyes. Polymer particles and micelles [38], liposomes [39], gold, calcium and silica nanoparticles [40,41,42] are especially popular among the nanoparticles used for the delivery of PSs. Additionally, nanoparticles can be modified with stimulus-sensitive moieties. Stimulus-sensitive carriers are able to respond to specific factors of the tumor microenvironment, which leads to the disintegration of the carrier in the tumor site and controlled release of the drug. The influencing factors can be of exogenous (magnetic field, heating, exposure to UV and IR radiation or ultrasound) [32], and endogenous nature—changes in the pH level [41], enzymes and redox potential [43].

Earlier, we described the novel group of tetracyanotetra(aryl)porphyrazines as prospective photosensitizers for PDT [44,45,46,47]. The uniqueness of these compounds lies in the fact that they combine the properties of effective photodynamic agents and microenvironment sensors [48,49].

We have previously shown that cyanoporphyrazines can be efficiently loaded in nanoscale and submicron containers of different natures. The encapsulation of hydrophobic cyanoporphyrazine dye into liposomes of different lipid composition promotes their high uptake by tumor cells while preserving the photoinduced toxicity of the PS [50]. The dyes of the cyanoporphyrazine group were also successfully loaded into amphiphilic polymer brushes and, due to the presence of such a container, were selectively accumulated and retained in tumors, providing subsequent successful photodynamic therapy [49,51]. The potential of cyanoporphyrazines was fully shown in the example of vaterite particles [52]. The unique property of these dyes, namely, the dependence of the fluorescence lifetime on the microenvironment, allowed for the visualization of the porphyrazines’ release from the vaterite in real time in vivo. The obtained results aroused interest in the further modification and study of cyanoporphyrazines, not only as potential agents for PDT, but also as sensors to assess the dynamics of drug release from the transport carrier.

In this study, we perform a detailed analysis of two photosensitizers, namely, tetra(2-naphthyl)tetracyanoporphyrazine (hereinafter, **PzNPh**) and its iron complex (hereinafter, **FePzNPh**) in order to assess their possible application in PDT, and assessed the effect of an iron atom chelating by porphyrazine macrocycle. Interest in the iron complex is due to the fact that a number of studies have shown that the presence of an iron cation at the center of the PS macrocycle significantly changes its photochemical and photophysical characteristics, which, as a consequence, leads to a change in its photobiological properties [21,22]. Herein, we present data on the photophysical properties of the photosensitizers, including the absorption and fluorescence spectra; fluorescence quantum yield in low and high viscosity media; and the dynamics of photobleaching and photodegradation quantum yield. In addition, we study the cellular uptake and dark and photoinduced cytotoxicity of **PzNPh** and **FePzNPh** against cancer cells in culture.

## 2. Materials and Methods

### 2.1. Photosensitizers

Two photoactive dyes were tested: tetra(2-naphthyl)tetracyanoporphyrazine (hereinafter, **PzNPh**) and its iron complex (hereinafter, **FePzNPh**) (Figure 1).

The general scheme of the preparation of the free base **PzNPh** comprised the following stages: naphthyl tricyanoethylene synthesis, template assembly of the macrocycle at the Yb^3+^ cation, and elimination of the central cation to form the free base. The details of **PzNPh** preparation were described previously [44,53]. The product was carefully purified by chromatography (Silica gel 60, 40-µm, Merk, THF eluent). The purification was repeated at least four to six times.

For the **FePzNPh** preparation, the degassed ethanol (3 mL) was added to the mixture of FeAc_2_ (N-MP, 95%, Aldrich, St. Louis, MO, USA) (35 mg, 0.00023 mol) and **PzNPh** (19 mg, 0.000023 mol) in inert atmosphere and the solution was kept for about 24 h. Then, the solvent was removed under vacuum and the dark blue precipitate was washed with pure ethanol (5 mL, 5 portions) to remove acetic acid (Figure 1). Yield 80%.

### 2.2. Analytical Data on FePzNPh

IR spectra in mineral oil suspensions were recorded using an FSM 1201 spectrometer (Appendix A).

IR (KBr, ν_max_/cm^−1^): 2197 (C≡N); 1625 (shoulder), 1595 (shoulder), 1571, 1557, 1541, 1414 (C=N; C=C); 1300, 1261, 1215 (C_ar_–H, C=N); 1046, 1027, 970, 952 (C_ar_–H; C=C).

Positive ion electron ionization mass spectra were measured on a PolarisQ/TraceGCUltra GC/MS spectrometer (Waltham, MA, USA) at 70 eV in the mass number range of 50–1000 (Appendix A).

EI MS (70 eV): m/z (%) 369 [M^+^H_2_O-2Ph]^2+^ (35), 456 [M^+^H_2_O-3CN]^2+^ (100), 482 [M^+^H_2_O-CN]^2+^ (15), 504 [M^+^2H_2_O]^2+^ (22).

Analytical data for PzNPh has been presented previously [44].

### 2.3. Cell Culture

The experiments were performed using human epidermoid carcinoma cells A431 (All-Russian Collection of Cell Cultures, Institute of Cytology of the Russian Academy of Sciences, Saint-Petersburg, Russia), human keratinocyte cells HaCaT and human adenocarcinoma ovarian cells SKOV-3.ip (provided from the cell collection of the Institute of Bioorganic Chemistry at the Russian Academy of Sciences, Moscow, Russia). The cells were cultured in 25 cm^2^ culture flasks (Corning, Corning, NY, USA) at 37 °C under 5% CO_2_ in Dulbecco’s Modified Eagle Medium (DMEM, HyClone, Logan, UT, USA) containing 2 mM L-glutamine (PanEco, Moscow, Russia) and 10% fetal bovine serum (FBS, Thermo Fisher, Waltham, MA, USA). For passaging, the cells were carefully detached using a trypsin-versene solution (1:1) (PanEco, Moscow, Russia).

### 2.4. Analysis of Spectral Properties and Fluorescence Quantum Yield

To register the absorption and fluorescence spectra of **PzNPh** and **FePzNPh**, we used a Synergy MX plate reader (BioTek, Winooski, VT, USA). As a solvent, we used deionized water and glycerol, the concentration of **PzNPh** and **FePzNPh** in all studies was 5 µM. Absorption was recorded in the range of 300–700 nm, fluorescence was recorded in the range of 600–800 nm with excitation at a wavelength of 570 nm.

The quantum yield of fluorescence was determined relative to rhodamine B in ethanol (*φ* = 0.7) [54]:(1)$$\varphi_{1}=\frac{\varphi_{0}\times F_{1}\times D_{0}^{}}{F_{0}\times D_{1}}$$
where *φ_0_* is the quantum yield of rhodamine; *φ*_1_ is the quantum yield of the tested PS; *F*_1_ and *F*_0_ are the area under the fluorescence curve for the tested PS (**PzNPh** or **FePzNPh**) and rhodamine B, respectively; *D*_1_ and *D*_0_ are the optical density of the PS and rhodamine B at the excitation wavelength.

### 2.5. Photobleaching

Photobleaching was analyzed for the **PzNPh** or **FePzNPh** solutions in glycerol at a concentration of 5 µM. When analyzing the photobleaching, the solutions were placed in the wells of a 96-well plate and irradiated with an LED light source for microplates (λ_ex_ 615–635 nm, 20 mW/cm^2^) [55]. Registration of the absorption spectra was carried out in the wavelength range of 450–700 nm before and after irradiation in various light doses. The measurements were taken using a Synergy MX plate reader. The experiment was carried out at a room temperature of 25 °C.

The photodegradation quantum yield was calculated using the following equation [56,57]:(2)$$\varphi_{ph}=\frac{\frac{D_{0}-D}{D_{0}}\times C\times N_{A}\times V}{N_{q}}$$
where *D*_0_ and *D* are the optical density of the PS before and after irradiation; *C* is the concentration of the photosensitizer before irradiation; *N_A_* is the Avogadro constant; *V* is the volume of the solution and *N_q_* is the number of quanta absorbed by the solution.

*N_q_* was defined as:(3)$$N_{q}=\frac{(1-10^{-D})\times I\times \lambda \times t}{\hbar \times c}$$
where *I* is the radiation power density; *λ* is the wavelength of the irradiation light; *t* is the irradiation time in seconds; *c* is the speed of light; and *ℏ* is Planck’s constant.

### 2.6. Study of the Dynamics of Cellular Uptake

The A431 cells were planted in a 96-well plate at 10,000 cells per well and cultured for 12 h. Then, the medium was replaced with a fresh serum-free growth medium containing **PzNPh** or **FePzNPh** at a concentration of 5 µM. The accumulation of pophyrazines by cells was studied using laser scanning fluorescence confocal microscopy. We used a scanning confocal microscope Axio Observer Z1 LSM 710 NLO/Duo (Carl Zeiss, Jena, Germany) equipped with a C-Apochromat 63× water immersion objective lens with a numerical aperture of 1.2. Fluorescence was recorded in the range of 600–750 nm with excitation at a wavelength of 594 nm. Image analysis was performed before and at various time points after adding **PzNPh** and **FePzNPh** to the cells. The obtained images were used to analyze the fluorescence intensity of the porphyrazines in the cell cytoplasm. For this, the ZEN 2012 software (Carl Zeiss, Jena, Germany) was used; about 10 cells per field of view were used for the analysis.

To analyze the dynamics of cellular uptake, cells were planted in 96-well plates (Corning, Corning, NY, USA) in the amount of 10,000 cells per well and cultured overnight until they were completely attached. Then, the culture medium was replaced with a serum-free medium containing **PzNPh** or **FePzNPh** at a concentration of 5 μM, and the fluorescence intensity was recorded with a Synergy MX plate reader at a wavelength of 660 nm with excitation at 590 nm. The fluorescence signal was normalized to the maximum value recorded at the end of the incubation to eliminate the effect of cell culture density on the result.

### 2.7. Analysis of Dark Toxicity and Photodynamic Activity

The A431, HaCaT and SKOV-3.ip cells were planted in a 96-well plate at a density of 4000 cells for A431 and 2000 cells for HaCaT and SKOV-3.ip per well, respectively, and cultured for 12 h. Then, **PzNPh** or **FePzNPh** was added to the cells in a serum-free medium at a concentration range from 0.0001 to 100 μM and incubated for 4 h. After that, the medium with the porphyrazines was replaced with a complete culture medium.

To analyze the photodynamic activity, the cells were exposed to light at a dose of 20 J/cm^2^ using an LED light source (λ_ex_ 615–635 nm, 20 mW/cm^2^) [55]. To analyze dark toxicity, the cells were incubated with **PzNPh** or **FePzNPh** in the dark under CO_2_ incubator conditions for the same period of time. As a control group, we used cells that were incubated according to the same protocol, but without adding **PzNPh** or **FePzNPh**. The MTT test was carried out 24 h after light exposure.

To do this, the medium in the plates was replaced with a serum-free fresh growth medium with 0.5 mg/mL of MTT-reagent and incubated for 4 h. (Alfa Aesar, Lancashire, UK). After that, the medium was taken and dimethyl sulfoxide (DMSO, PanEco, Moscow, Russia) was added, 200 μL per well, to dissolve the formed formazan crystals. The optical density of the resulting solutions at a wavelength of 570 nm was measured using a Synergy MX plate reader.

Based on the results, the cell viability was analyzed after photodynamic exposure in relation to the optical density of the formazan solution in each well to the control group. The resulting dose-effect relationships were used to calculate the IC_50_, the dose of the test substance leading to inhibition of cell culture growth by 50%. Each experiment was performed in triplicate.

We also conducted a study on the production of reactive oxygen species by a chemical trap method using 1,3-diphenylisobenzofuran (DPBP) at a concentration of 200 μM in an ethanol-glycerol mixture (50%). Combinations of **PzNPh** and **FePzNPh** 5 μM were used for the measurements. The solutions were irradiated with light at a dose of 0 to 20 J/cm^2^ using an LED light source (λ_ex_ 615–635 nm, 20 mW/cm^2^). The absorbance of solutions at a wavelength of 420 nm was measured using a Synergy MX plate reader.

### 2.8. Statistical Analysis

Statistical analysis was performed in GraphPad Prism (v.9.0) (GraphPad Software, San Diego, CA, USA). Cell death was analyzed by ANOVA followed by Dunnett’s test.

## 3. Results

Two types of photoactive dyes, tetra(2-naphthyl)tetracyanoporphyrazine (**PzNPh**) and its iron complex (**FePzNPh**) were studied (Figure 1). The cyanoporphyrazine iron complex **FePzNPh** was synthesized and studied for the first time. Despite the previously reported properties of **PzNPh [44]**, the characteristics of the individual batches of the compounds can significantly differ. Thus, we repeated the spectral and cell studies to prove the preservation of the compound’s properties between the synthesis procedures.

### 3.1. Spectral Properties and Fluorescence Quantum Yield of PzNPh and FePzNPh

The studied compounds belong to the cyanoporphyrazine group, for which the dependence of fluorescent properties, such as quantum yield and fluorescence lifetime, on the viscosity of the medium is shown [48]. Assuming the rotary properties of **PzNPh** and **FePzNPh**, we have studied the spectral properties of both water and glycerol. It should be noted that the measurements in glycerol were carried out at a temperature of 25 °C; the viscosity of the glycerol was 858.4 cP.

The analysis of the photophysical properties of PSs has revealed that both porphyrazines absorb and fluoresce in red and far-red regions of the spectrum 500–650 nm and 600–800 nm, respectively (Figure 2, Table 1) [44].

We have registered an increase fluorescence quantum yield for **PzNPh** and **FePzNPh** in glycerol as compared to an aqueous solution, which confirms our assumption that the studied compounds belong to the class of fluorescent molecular rotors. Thus, the quantum yield of **PzNPh** in glycerol increased 21.5 times compared to an aqueous solution, and 10.4 times for **FePzNPh**. The presence of an iron atom led to an increase in the quantum yield of fluorescence both in water and in glycerol by 3 and 1.5 times, respectively (Table 1).

The presence of iron in the porphyrazine macrocycle has led to a bathochromic shift of the absorption peak by 18 nm when water was used as a solvent [44]; while in glycerol, the position of the absorption maxima for **PzNPh** and **FePzNPh** was the same (Figure 2, Table 1). The addition of an iron atom has also led to a decrease in the molar extinction coefficient by a factor of 1.5 in water and by a factor of 2.6 in glycerol.

Moreover, the presence of iron has not affected the positions of the fluorescence maxima in glycerol. However, an increase in viscosity dissolved the shift of the fluorescence peak to the short wavelength region of the spectrum (Table 1). It should be noted that the fluorescence peak of the studied porphyrazines is quite wide compared to the “traditional” PSs, which is caused by the peculiarities of their photophysical properties as fluorescent molecular rotors. They are characterized by the twisted intramolecular charge-transfer (TICT) state, which is accompanied by the formation of TICT-conformers, whose fluorescent signals are shifted to the long-wavelength region, which leads to a broadening of the fluorescence spectrum [58].

### 3.2. Photobleaching of PzNPh and FePzNPh

The photodegradation quantum yield is an important parameter when calculating the dosage of a photosensitizer used in photodynamic therapy. The high photostability of the photosensitizer will avoid the use of high doses of the PS and reduce the radiation dose required to achieve a similar clinical result when using low-stability PSs [56].

To estimate the photobleaching coefficient of photosensitizers, we used a solution of **PzNPh** and **FePzNPh** in glycerol. Glycerol was used as a solvent in the assessment of the photobleaching because of the photophysical features of the studied dyes. When energy is transferred to them in the form of a quantum of light, the resulting energy of the excited state can be spent through three main channels—intramolecular rotation, fluorescence, non-radiative internal conversion, singlet oxygen generation, and thermal relaxation. In low-viscosity media, the priority channel for energy consumption is the rotation of separate parts of the molecule. The use of a relatively viscous medium reduces the contribution of the rotor movement and increases the contribution of other processes. It is important to note that such conditions are more relevant when modeling the behavior of the PS in the intracellular environment during the photodynamic reaction [48].

Both compounds demonstrated rather high photostability: at a dose of 100 J/cm^2^, the percentage of photobleaching was about 50%. The addition of an iron atom into the macrocycle led to an increase in the photostability of porphyrazine, so the photodegradation quantum yield reached 5.7 × 10^−7^ for **PzNPh** and 4 × 10^−7^ for **FePzNPh** (Figure 3).

### 3.3. Cellular Uptake of PzNPh and FePzNPh

The photobiological properties of **PzNPh** and **FePzNPh** were analyzed using human epidermoid carcinoma A431 cells. The choice of the cell line was due to the fact that the main area of PDT application is superficial tumors, including skin cancer [59,60]. Both photosensitizers were accumulated by cells, but with different dynamics: 15 min after adding the PSs to the incubation medium, the intensity of the fluorescence signal for **FePzNPh** was 2.5 times lower compared to **PzNPh**; but after 4 h of incubation, the fluorescence signals almost leveled out, which allowed us to assume a difference in the cellular uptake rates of **PzNPh** and **FePzNPh** (Figure 4). However, since the brightness of the fluorescence of **PzNPh** and **FePzNPh** differ even at the same viscosity values (Figure 2), we cannot make an absolute assessment regarding the concentration of the studied dyes in the cell.

To test the hypothesis concerning the difference in uptake rate for the two compounds, we studied the accumulation dynamics of porphyrazines by spectrophotometry (Figure 5). It should be noted that the unique photophysical properties of the studied porphyrazines make it possible to use a measurement technique based on continuous recording of the cell fluorescence signal using a plate spectrofluorometer. Porphyrazine added to the culture medium gives practically no fluorescence signal due to the low viscosity of the medium. When porphyrazine enters a cell with a much higher viscosity, the fluorescent signal of porphyrazine increases and, therefore, we can record its accumulation in the cell at different time points without changing the medium.

It has been shown that the introduction of iron leads to a decrease in the rate of entry of porphyrazine into the cell: the maximum accumulation for **PzNPh** was recorded only 1 h after its addition to the incubation medium, while for **FePzNPh**, the maximum was reached after 4 h (Figure 5). This may be caused by the interaction of iron with negatively charged groups of phospholipid bilayers. The results obtained confirm and explain the data recorded by laser scanning confocal microscopy. The results are consistent with those obtained earlier for **PzNPh** [44].

### 3.4. Dark Toxicity and Photodynamic Activity of PzNPh u FePzNPh

We analyzed the toxic effects of **PzNPh** and **FePzNPh** against the A431 cells. The cells were pre-incubated with the dyes for 4 h, which corresponds to the reaching of their maximum cellular accumulation. Both compounds showed high photodynamic activity when exposed to light (Figure 6). The inclusion of an iron atom had a pronounced effect on the toxic properties. The photodynamic activity decreased by about two times, with an IC_50_ value of 190 nM for **PzNPh** and 400 nM for **FePzNPh** (Table 2). A study of the generation of reactive oxygen species (Appendix A) showed a higher generation of reactive oxygen for **PzNPh**, which is consistent with the data presented above. However, at the same time, we found a 3.5-fold decrease in dark toxicity, which led to an increase in the photodynamic index (PDI). For **FePzNPh**, it was significantly higher and reached a value of 60.

Data on cytotoxicity for **PzNPh** did not differ from those obtained previously [44]. This demonstrates a high degree of reproducibility of the synthesis procedure and is important for further practical implementation.

The cytotoxicity and photodynamic activity of both compounds were additionally analyzed against two cell lines of different origin: HaCaT human keratinocytes and SKOV-3.ip human ovarian adenocarcinoma (Figure 6, Table 2). In fact, inclusion of an iron atom led to varying effects depending of the cell line. We explain this by the metabolic peculiarities of the selected cell lines due to their different origin. However, for both cell lines, we observed an increase in the photodynamic index, similarly to A431. For HaCaT cells, it increased by about 6-fold and for SKOV-3.ip by 3.7-fold compared to the free base.

Despite the slight decrease in photoinduced toxicity for **FePzNPh** against A431 cells, the increase in the photodynamic index could potentially indicate a reduction in possible side-effects impacting healthy organs and tissues, and makes the iron complex very interesting for further study.

## 4. Discussion

The current development strategy concerning the concept of an ideal PS is aimed at making PDT a more personalized therapeutic approach, as well as ensuring the activation of an antitumor immune response, in addition to the elimination of cells in the primary focus tumor [61].

Within this trend, we have synthesized a novel group of porphyrazines as prospective photosensitizers for PDT [44,45,46,47]. The advantage of the investigated porphyrazines in comparison with the well-known clinically approved PSs lies in their additional modality to the photodynamic activity—the sensitivity of fluorescent properties to the viscous microenvironment. Porphyrazines belong to the class of molecular rotors, the photophysical properties (fluorescence quantum yield and lifetime) of which change significantly depending on the viscosity of the environment. In less viscous media, the excitation energy of porphyrazines is spent on intramolecular twisting or rotation of the side aryl groups, which leads to a very high nonradiative relaxation constant and, consequently, to a low fluorescence quantum yield and lifetime. In high viscosity media, intramolecular rotation is difficult, and therefore, there is a multiple increase in fluorescence quantum yield and lifetime. The high viscosity sensitivity of the photophysical properties of porphyrazines makes it possible to carry out dosimetric control of PDT and select treatment regimens for each patient [48,61]. Moreover, we have previously shown that some compounds from the group of porphyrazines are capable of inducing immunogenic cell death and can be used as effective photodynamic agents [20].

Our previous research indicates that porphyrazines are promising agents that can be used in personalized photodynamic therapy. Therefore, we continue to search for new promising compounds from this group with improved photophysical and photobiological properties.

In this article, we report on the comparative analysis of tetra(2-naphthyl)tetracyanoporphyrazine and its iron complex. We believe that the efficiency of PDT using **FePzNPh** as a PS can be significantly increased by iron(II) cations reversibly bound to the macrocycle. Such an effect in the presence of iron can be expected as a result of the development of the Fenton reaction (the formation of an hydroxide and hydroxyl radical by a reaction between Fe^2+^ and hydrogen peroxide) [61]. Our interest was also caused by recent publications on the role of iron-catalyzed enhancement of the antitumor effect due to the induction of ferroptosis as the cause of immunogenic cell death, which can effectively suppress tumor resistance to therapeutic effects [61,62].

**PzNPh** and **FePzNPh** absorb and fluoresce in red and far-red regions of the spectrum 500–650 nm and 600–800 nm, respectively. The presence of iron does not affect the shape of the spectra and the position of the absorption and fluorescence maxima recorded at a wavelength of 600 and 660 nm, respectively (Figure 3). In addition, we have registered an increase in the fluorescence quantum yield of **PzNPh** and **FePzNPh** in glycerol compared to an aqueous solution because this group of compounds belongs to the class of fluorescent molecular rotors.

In the presence of iron, the quantum yield increased by 3 and 1.5 times, in water and glycerol, respectively (Table 1). Iron has increased the photostability of porphyrazine, so the photobleaching quantum yield has reached 5.7 × 10^−7^ for **PzNPh** and 4 × 10^−7^ for **FePzNPh** (Figure 3).

A study performed on epidermoid carcinoma cells A431 has shown that **PzNPh** and **FePzNPh** are infiltrated and accumulated by tumor cells (Figure 5). The presence of iron in the macrocycle leads to a decrease in the rate of porphyrazine accumulation, which may be caused by the fact that iron interacts with negatively charged groups of phospholipid bilayers. This property can be of great importance for the pharmacokinetics of a drug in an in vivo system. Once in the blood, a PS is in two fractions: bound to plasma proteins or high- and low-density lipoproteins or bound to cellular components of the blood. Protein binding reduces PS diffusion into cells and tissues, while binding to intracellular components increases it, thereby, reducing the toxic load on non-target organs and tissues.

Analysis of the photodynamic activity has shown that the addition of an iron cation to the macrocycle leads to an increase in the photodynamic index (Table 2).

Thus, the photodynamic index of **FePzNPh** as compared with **PzNPh** in the experiment on A431 almost doubled. The increase was 6.1 times for HaCaT, and 3.7 times for SKOV-3.ip. We can explain such differences by the difference in metabolism of the chosen cell lines. However, in all the cases, the photodynamic index of a metal complex was higher than that of a metal-free base.

The results obtained in this study play an important role for clinical PDT, since a high photodynamic index may potentially indicate a decrease in possible side effects impacting healthy organs and tissues.

To sum up, our studies reveal that **PzNPh** and **FePzNPh** could be considered as potent agents for PDT with promising photophysical and photobiological properties.

## 5. Conclusions

We conducted a comparative analysis of the photophysical and photobiological properties of **PzNPh** and newly synthesized **FePzNPh** and studied the influence of iron as a central cation in the porphyrazine macrocycle. The resulting compounds have spectral properties suitable for PDT. In particular, they are characterized by absorption and fluorescence in red and far-red regions of the spectrum of tissues and are characterized by the sensitivity of the fluorescent properties to the viscosity of the medium. We have shown that iron has a significant effect on the photophysical properties of cyanoporphyrazine: in the presence of iron, the quantum yield of fluorescence and the photostability of porphyrazine increases. A study conducted on epidermoid carcinoma cells has shown that **PzNPh** and **FePzNPh** penetrate tumor cells and are accumulated by these cells, while the presence of an iron atom leads to a decrease in the rate of accumulation of the compound by cells, which may be related to the fact that iron interacts with negatively charged groups of phospholipid bilayers. This feature of **FePzNPh** may be of further importance for the pharmacokinetics of the compound in in vivo experiments. The compounds show high photodynamic activity against tumor cells in vitro. In addition, the presence of the iron cation leads to an increase in the photodynamic index and a decrease in dark toxicity, which could potentially indicate a decrease in side effects for healthy organs and tissues at the level of the whole organism. In general, both compounds are promising for PDT and can be recommended for further in vivo studies.

## Figures and Tables

**Figure 1 pharmaceutics-14-02655-f001:**
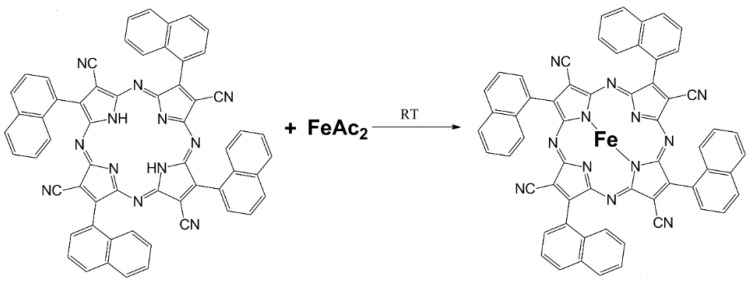
Scheme for the synthesis of **FePzNPh** from **PzNPh**.

**Figure 2 pharmaceutics-14-02655-f002:**
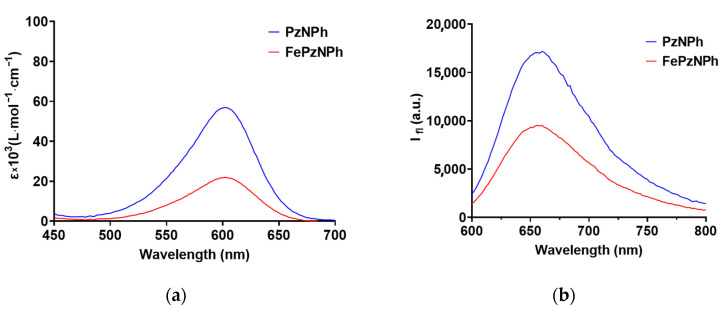
Absorption and fluorescence spectra of **PzNPh** and **FePzNPh** in glycerol (**a**,**b**) and in water (**c**,**d**) (both at 5 µM). The fluorescence was excited at λ_ex_ 570 nm.

**Figure 3 pharmaceutics-14-02655-f003:**
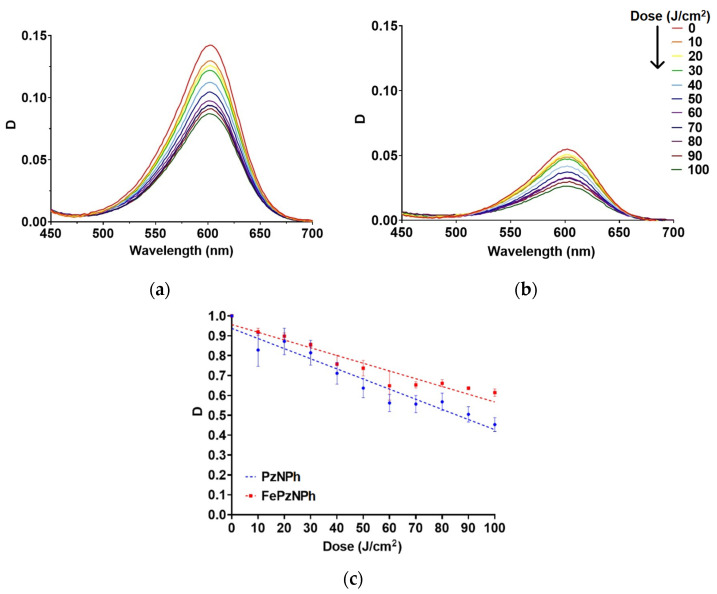
Photobleaching of **PzNPh** and **FePzNPh** in the glycerol solution under irradiation; (**a**)—Absorption spectra of **PzNPh** after irradiation; (**b**)—Absorption spectra of **FePzNPh** after irradiation; (**c**)—The dose-dependent decrease in optical density at the absorption maximum for **PzNPh** and **FePzNPh**. The data are presented as the mean values ± SD (*n* = 3).

**Figure 4 pharmaceutics-14-02655-f004:**
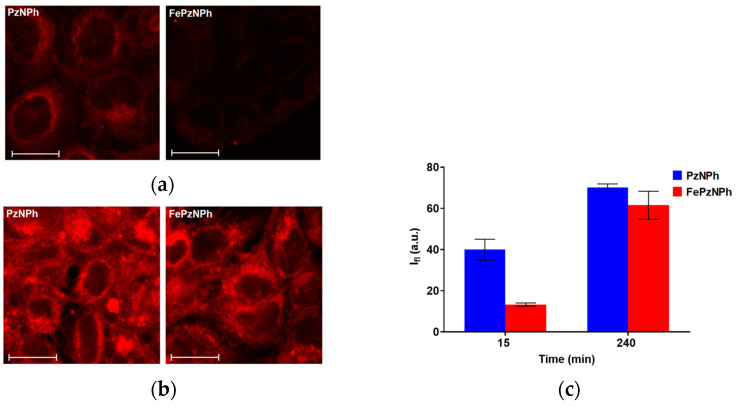
Analysis of the cellular uptake of **PzNPh** and **FePzNPh**. Representative confocal images of human epidermoid carcinoma A431 cells after 15 min (**a**) and 240 min (**b**) of incubation with **PzNPh** and **FePzNPh** (5 µM). Relative accumulation of **PzNPh** and **FePzNPh** after 15 and 240 min incubation, estimated from a confocal microscopy experiment (**c**) (*n* ≥ 10). The values are given minus the background, which did not exceed 4 a.u. Scale bar 20 µm. The data are presented as the mean values ± SD.

**Figure 5 pharmaceutics-14-02655-f005:**
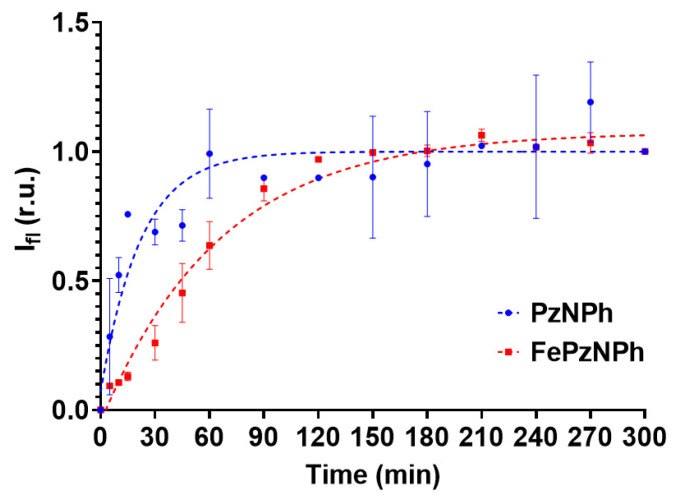
Dynamics of **PzNPh** and **FePzNPh** accumulation in human epidermoid adenocarcinoma A431 cells. The fluorescence signal was normalized to the maximum value recorded at the end of the incubation period to eliminate the effect of cell culture density on the result. The data are presented as the mean values ± SD (*n* = 3).

**Figure 6 pharmaceutics-14-02655-f006:**
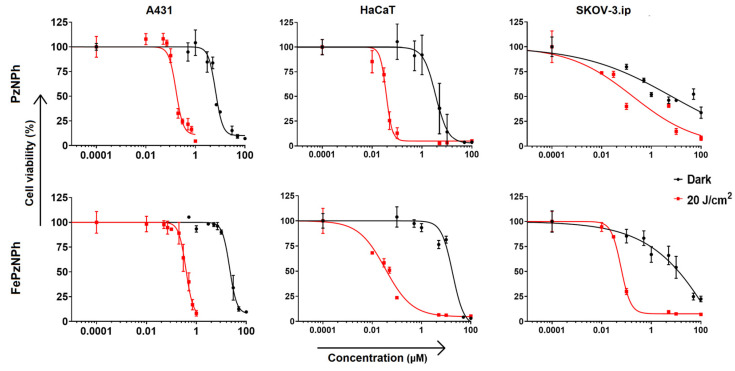
Dark toxicity and photodynamic activity of **PzNPh** and **FePzNPh** against human epidermoid carcinoma A431 cells, human keratinocytes HaCaT and human ovarian adenocarcinoma SKOV-3.ip cells. To induce a photodynamic response, the cells were irradiated with light at a dose of 20 J/cm^2^ (615–635 nm, 20 mW/cm^2^). The data are presented as the mean values ± SD (*n* = 3).

**Table 1 pharmaceutics-14-02655-t001:** Spectral properties and fluorescence quantum yield of **PzNPh** and **FePzNPh**.

Compound	H_2_O				Glycerol			
	λ_abs_ (nm)	λ_em_ (nm)	ε, (L·mol^−1^·cm^−1^)	φ (%)	λ_abs_ (nm)	λ_em_ (nm)	ε, (L·mol^−1^·cm^−1^)	φ (%)
**PzNPh**	592 *	690 *	3.5 × 10^4^ *	0.6	600	660	5.7 × 10^4^	12.9
**FePzNPh**	610	675	2.4 × 10^4^	1.8	600	656	2.2 × 10^4^	18.7

* data obtained earlier in the article [44].

**Table 2 pharmaceutics-14-02655-t002:** Dark toxicity and photodynamic activity of **PzNPh** and **FePzNPh** against A431, HaCaT, and SKOV-3.ip cells.

Cell Line	Compound	* IC_50_ Dark (μM)	* IC_50_ Light (nM)	Photodynamic Index
A431	**PzNPh**	7.04 (6.13–8.16)	190 (160–210)	37
	**FePzNPh**	24.18 (22.03–26.35)	400 (370–440)	60
HaCaT	**PzNPh**	3.53 (2.22–5.10)	40 (30–40)	88
	**FePzNPh**	18.33 (13.61–29.52)	34 (20–40)	539
SKOV-3.ip	**PzNPh**	6.61 (3.088–15,84)	180 (71–460)	37
	**FePzNPh**	9.6 (6.37–14.63)	70 (60–80)	137

* mean IC_50_ values and 95% confidence interval are indicated.

## Data Availability

The data used to support the findings of this study are available from the corresponding author upon request.

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
