# Peer review of "Comparative Analysis of Tetra(2-naphthyl)tetracyano-porphyrazine and Its Iron Complex as Photosensitizers for Anticancer Photodynamic Therapy"

_pharmaceutics, 2022, doi:10.3390/pharmaceutics14122655_

Round 1

Reviewer 1 Report

The manuscript describes the investigation of novel photosensitizer (PS) as anti-cancer agents. The authors discuss the possibility to enrich these PS in the cells, investigate cellular death upon light irradiation and dark states and investigate also photostability. The paper is very interesting and meaningful, but requires a revision prior to publication.

1.     Please write also full word and abbreviation upon first use of photodynamic therapy.

2.    Novel particulate (adsorbed dyes)[1], dispersed[2,3], or remote steerable containers[4] could more precisely deliver the PS to the target. The authors should mention such extended future uses. This could reduce common side effects like blindness if the PS migrates to the eyes. The µm size makes investigations of their fate easier and easier steerable.

3.     Equation 1-3 is unreasonably large, please reduce size to match the text size for the symbols.

4.     Page 1 line 143 symbol error.

5.     Where did the authors get the cells from?

6.     Did the authors synthesize their PS themselves? They mention it, where are the source chemicals for this synthesis from? Where is the reference to this synthesis route or a detailed description?

7.     What was the purity of the PS?

8.     Optimum window of biological tissue is as far as I know in the range of 650-1100. It is better than shorter wavelengths, so the authors should not exaggerate their claim. Suitable window is enough.

References

[1]      S. Rutkowski, L. Mu, T. Si, M. Gai, M. Sun, J. Frueh, Q. He, Magnetically-propelled hydrogel particle motors produced by ultrasound assisted hydrodynamic electrospray ionization jetting, Colloids Surfaces B Biointerfaces. 175 (2019) 44–55. https://doi.org/10.1016/j.colsurfb.2018.11.068.

[2]      K. Köhler, H. Möhwald, G.B. Sukhorukov, Thermal behavior of polyelectrolyte multilayer microcapsules: 2. Insight into molecular mechanisms for the PDADMAC/PSS system, J. Phys. Chem. B. 110 (2006) 24002–10. https://doi.org/10.1021/jp062907a.

[3]     K. Köhler, G.B. Sukhorukov, Heat treatment of polyelectrolyte multilayer capsules: A versatile method for encapsulation, Adv Funct Mater. 17 (2007) 2053–2061. https://doi.org/10.1002/adfm.200600593.

[4]      S. Rutkowski, T. Si, M. Gai, M. Sun, J. Frueh, Q. He, Magnetically-guided Hydrogel Capsule Motors produced via Ultrasound assisted Hydrodynamic Electrospray Ionization Jetting, J. Colloid Interface Sci. 25 (2019) 752–774. https://doi.org/10.1016/j.jcis.2019.01.103.

Author Response

Dear Reviewer,

We would like to express our sincere appreciation for your careful attention to our manuscript and for the suggested improvements and valuable comments. We have revised the manuscript according to your remarks. Please find below the detailed description of the revision with our point-by-point answers.

Reviewer: The manuscript describes the investigation of novel photosensitizer (PS) as anticancer agents. The authors discuss the possibility to enrich these PS in the cells, investigate cellular death upon light irradiation and dark states and investigate also photostability. The paper is very interesting and meaningful, but requires a revision prior to publication.

Point 1: Please write also full word and abbrevi ation upon first use of photodynamic therapy.

Response 1: We added the acronym PDT at first mention of photodynamic therapy in the Introduction section (Line 33 in the revised version of the manuscript).

Quotation: Photodynamic therapy (PDT) was developed over forty years ago and is now widely is used to treat of superficial tumors or tumors accessible with an endoscope.

Point 2: Novel particulate (adsorbed dyes) [1], dispersed [2,3], or remote steerable containers [4] could more precisely deliver the PS to the target. The authors should mention such extended future uses. This could reduce common side effects like blindness if the PS migrates to the eyes. The µm size makes investigations of their fate easier and easier steerable.

References

[1] S. Rutkowski, L. Mu, T. Si, M. Gai, M. Sun, J. Frueh, Q. He, Magnetically-propelled hydrogel particle motors produced by ultrasound assisted hydrodynamic electrospray ionization jetting, Colloids Surfaces B Biointerfaces. 175 (2019) 44–55. https://doi.org/10.1016/j.colsurfb.2018.11.068.

[2] K. Köhler, H. Möhwald, G.B. Sukhorukov, Thermal behavior of polyelectrolyte multilayer microcapsules: 2. Insight into molecular mechanisms for the PDADMAC/PSS system, J. Phys. Chem. B. 110 (2006) 24002–10. https://doi.org/10.1021/jp062907a.

[3] K. Köhler, G.B. Sukhorukov, Heat treatment of polyelectrolyte multilayer capsules: A versatile method for encapsulation, Adv Funct Mater. 17 (2007) 2053–2061. https://doi.org/10.1002/adfm.200600593.

[4] S. Rutkowski, T. Si, M. Gai, M. Sun, J. Frueh, Q. He, Magnetically-guided Hydrogel Capsule Motors produced via Ultrasound assisted Hydrodynamic Electrospray Ionization Jetting, J. Colloid Interface Sci. 25 (2019) 752–774. https://doi.org/10.1016/j.jcis.2019.01.103.

Response 2: We added the discussion on the topic drug delivery systems for photosensitizers to the Introduction section with links to the recommended articles as well as some additional relevant sources (Lines 80-93 in the revised version of the manuscript).

Quotation: Another approach to increasing the effectiveness of PDT is the development of delivery systems for photosensitizers [32]. Nanoscale carriers are of particular interest as delivery systems. They can provide an enhancement in PS solubility and prolonged blood circulation of PS, facilitate PS targeted delivery and controlled release in the pathological site [33-37]. Targeted accumulation of photosensitizers in the tumor helps to reduce the serious side effects of PDT, such as prolonged (over several months) photosensitivity of the skin and eyes . Polymer particles and micelles [38], liposomes [39], gold, calcium and silica nanoparticles [40-42] are especially popular among the nanoparticles used for the delivery of PSs. Additionally, nanoparticles can be modified with stimulus-sensitive moieties. Stimulus-sensitive carriers are able to respond to specific factors of tumor microenvironment which leads to the disintegration of the carrier in the tumor site and controlled release of the drug. The influencing factors can be of exogenous (magnetic field, heating, exposure to UV and IR radiation or ultrasound) [32] and endogenous nature – changes in the pH level [41], enzymes and redox potential [43].

Point 3: Equation 1-3 is unreasonably large, please reduce size to match the text size for the symbols.

Response 3: We reduced the size of Equations 1-3 as recommended.

Point 4: Page 1 line 143 symbol error.

Response 4: Corrected

Point 5: Where did the authors get the cells from?

Response 5: The origin of the cell cultures is added to the manuscript (Lines 154-157 in the revised version of the manuscript).

Quotation: The experiments were performed using human epidermoid carcinoma cells A431 (All-Russian Collection of Cell Cultures, Institute of Cytology of the Russian Academy of Sciences, Saint-Petersburg, Russia), human keratinocyte cells HaСat and human adenocarcinoma ovarian cells SKOV-3.ip (provided from cell collection of the Institute of Bioorganic Chemistry of the Russian Academy of Sciences, Moscow, Russia). 

Point 6: Did the authors synthesize their PS themselves? They mention it, where are the source chemicals for this synthesis from? Where is the reference to this synthesis route or a detailed description?

Response 6: The synthesis of PS had been performed by the authors of this paper. We have added a detailed description of the synthesis of PzNPh and FePzNPh  and indicated the sources of chemicals for their synthesis (Lines 129-139 in the revised version of the manuscript).

Point 7: What was the purity of the PS?

Response 7: The product of PzNPh synthesis was carefully purified by chromatography (Silica gel 60, 40-µm, Merk, THF eluent). Purification was repeated at least four to six times.

We don’t subject  the prepared FePzNPh to the purification with silica since (i) we used carefully purified free base for its preparation; (ii) only undesirable impurity (AcOH) was removed by washing of FePzNPh precipitate with pure ethanol (5 ml, 5 portions) that had been removed under vacuum afterwards.

We added information about PzNPh and FePzNPh purification to the manuscript (Lines 133-139 in the revised version of the manuscript).

Point 8: Optimum window of biological tissue is as far as I know in the range of 650-1100. It is better than shorter wavelengths, so the authors should not exaggerate their claim. Suitable window is enough.

Response 8: We agree with the reviewer that in most literature sources the range of the optical transparency window is indicated as 650-1100 nm. However, there are number of works where, according to the authors, the range of the optical transparency window is 600-1000 nm [1-3].  Nevertheless, in order to avoid ambiguity, we have removed the term "optical transparency window of biological tissues" from the description of the photophysical properties of the porphyrazines under study.

References: 

  1. Richards-Kortum R, Sevick-Muraca E. Quantitative optical spectroscopy for tissue diagnosis. Annu Rev Phys Chem. 1996; 47:555-606. doi: 10.1146/annurev.physchem.47.1.555.
  2. Valery V. Tuchin Tissue Optics and Photonics: Light-Tissue Interaction II / doi: 10.18287/JBPE16.02.030201; Rocıo del Pilar Soto Astorga Haemoglobin sensing with optical spectroscopy during minimally invasive procedures// 2014, book
  3. Valentina Bello, Elisabetta Bodo, Sara Pizzurro and Sabina Merlo In Vivo Recognition of Vascular Structures by Near-Infrared Transillumination // doi:10.3390/ecsa-6-06639

Quotation: The analysis of photophysical properties of PSs has revealed that both porphyrazines absorb and fluoresce in red and far-red regions of the spectrum 500-650 nm and 600-800 nm, respectively (Line 259 in the revised version of the manuscript).

PzNPh and FePzNPh absorb and fluoresce in red and far-red regions of the spectrum 500-650 nm and 600-800 nm, respectively. (Line 421 in the revised version of the manuscript).

In particular, they are characterized by absorption and fluorescence in red and far-red regions of the spectrum of tissues and are characterized by the sensitivity of fluorescent properties to the viscosity of the medium. (Line 458 in the revised version of the manuscript).

Reviewer 2 Report

The manuscript describes the synthesis and PDT ability of new derivatives of porphyrazine. The study was planned well but still, some important data is missing as well as some inconsistencies within the presented results. My concerns are:

Check English, for ex.  Line 83- In the study [39] showed how, and following lines. Connective words are missing. A similar error has been found in several other places.

Figure 2: The dye concentration used for obtaining these spectra is missing.

It was reported that viscosity influences the photo-activity of dyes. Thus, how are the results in glycerol relevant to clinical testing? Further, in uptake studies, the author claims that no fluorescence was detected in culture media, however, the result in the water tells a different story.

The accumulation rate is slower for Fe complex, based on fluorescence intensity detected within cells. However, the fluorescence spectra already suggested that the Fe complex generates low fluorescence compared to non-modified compound. How has this been considered to determine the uptake?

The compound purity and structural characterization data are missing.

Physiochemical characteristics such as solubility in different mediums and stability are also relevant data if the aim is to utilize them for PDT.

Study missing to prove that these compounds release ROS, which is essential for PDT.

Author Response

Dear Reviewer,

We would like to express our sincere appreciation for your careful attention to our manuscript and for the suggested improvements and valuable comments. We have revised the manuscript according to your remarks. Please find below the detailed description of the revision with our point-by-point answers.

Reviewer: The manuscript describes the synthesis and PDT ability of new derivatives of porphyrazine. The study was planned well but still, some important data is missing as well as some inconsistencies within the presented results.

Point 1: Check English, for ex.  Line 83- In the study [39] showed how, and following lines. Connective words are missing. A similar error has been found in several other places.

Response 1: Corrected. In addition, we rely on editing by MDPI's translators.

Quotation: We have previously shown that cyanoporphyrazines can be efficiently loaded in nanoscale and submicron containers of different nature. Encapsulation of hydrophobic cyanoporphyrazine dye into liposomes of different lipid composition promoted their high uptake by tumor cells while preserving the photoinduced toxicity of the PS [50]. The dyes of cyanoporphyrazine group were also successfully loaded into amphiphilic polymer brushes and, due to the presence of such a container, were selectively accumulated and retained in tumors, providing subsequent successful photodynamic therapy [51],[49]. The potential of cyanoporphyrazines was fully shown on the example of vaterite particles [52]. The unique property of these dyes, namely, the dependence of the fluorescence lifetime on the microenvironment, allowed visualization of porphyrazines release from vaterite in real time in vivo. The obtained results arousing interest in further modification and study of cyanoporphyrazines not only as potential agents for PDT, but also as sensors to assess the dynamics of drug release from the transport carrier.

Point 2: Figure 2: The dye concentration used for obtaining these spectra is missing.

Response 2: Corrected. (Line 281 in the revised version of the manuscript).

Quotation: Figure 2. Absorption (a) and fluorescence (b) spectra of PzNPh and FePzNPh (both at 5 µM) in glycerol. The fluorescence was excited at λex 570 nm.

Point 3: It was reported that viscosity influences the photo-activity of dyes. Thus, how are the results in glycerol relevant to clinical testing? Further, in uptake studies, the author claims that no fluorescence was detected in culture media, however, the result in the water tells a different story.

Response 3: The studied compounds (PzNPh and FePzNPh) are characterized by the dependence of the lifetime of the excited state and the quantum yield of fluorescence on the viscosity of the medium. According to a number of literature sources, it is known that the viscosity values of the cytoplasm of living cells ranges from 2 сP to 100 сP [1, 2]. These values are corresponding with the viscosity range of glycerin-ethanol mixtures (~10-70% of glycerin concentration at room temperature) [3].

The fluorescence spectra of PzNPh and FePzNPh were studied using a high sensitivity (2 pM fluorescein typical) Synergy MX plate reader. This made it possible to get a very low fluorescence intensity emission spectra in water.

The accumulation of PzNPh and FePzNPh by cells was studied using a confocal microscope Axio Observer Z1 LSM 710 NLO/Duo (Carl Zeiss, Germany). Its detector sensitivity is much lower than the Synergy MX plate reader in use. So, in cells experiment, the fluorescence signal of both PzNPh and FePzNPh solutions could not be distinguished from electronic noise in the growth medium.

References:

  1. Fushimi K, Verkman AS. Low viscosity in the aqueous domain of cell cytoplasm measured by picosecond polarization microfluorimetry. J Cell Biol. 1991 Feb;112(4):719-25. doi: 10.1083/jcb.112.4.719
  2. Luby-Phelps K, Mujumdar S, Mujumdar RB, Ernst LA, Galbraith W, Waggoner AS. A novel fluorescence ratiometric method confirms the low solvent viscosity of the cytoplasm. Biophys J. 1993 Jul;65(1):236-42. doi: 10.1016/S0006-3495(93)81075-0
  3. J. B. Segur and Helen E. Oberstar Viscosity of Glycerol and Its Aqueous Solutions Ind. Eng. Chem. 1951, 43, 9, 2117–2120

Point 4: The accumulation rate is slower for Fe complex, based on fluorescence intensity detected within cells. However, the fluorescence spectra already suggested that the Fe complex generates low fluorescence compared to non-modified compound. How has this been considered to determine the uptake?

Response 4: In this study, we assessed the rate of reaching the maximum possible brightness value, but not the absolute concentrations of photosensitizers in the cell. This is due to the dependence of the fluorescence quantum yield on the viscosity of the medium. To avoid misinterpretation of the results by readers, we have added the statement into the article text. (Lines 321-324 in the revised version of the manuscript).

Quotation: However, since the fluorescence brightnesses of PzNPh and FePzNPh differ even at the same viscosity values (Figure 2), we cannot make absolute estimates regarding the concentration of the studied dyes in the cell.

Point 5: The compound purity and structural characterization data are missing.

Response 5: We have added data on the purity of the compounds and their structural characterization. (Lines 133-139; 141-150 in the revised version of the manuscript).

Point 6: Physiochemical characteristics such as solubility in different mediums and stability are also relevant data if the aim is to utilize them for PDT.

Response 6: PzNPh and FePzNPh are characterized by low solubility in water and glycerol. PzNPh shows good solubility in acetonitrile and tetrahydrofuran, but little solubility in toluene. As for FePzNPh, it is rather slightly soluble in all the solvents listed above. Both compounds show high solubility in dimethylsulfoxid (DMSO). In this regard, the stock aqueous solution for research contained about 20 wt.% DMSO and was further diluted by deionized water for the experiments. After preparing the compounds and before adding them to cell cultures, we controlled the solubility and retention of the compounds by the shape of the absorption and fluorescence spectra. No sighs of aggregation was detected in these conditions.
Since no quantitative measurement of PzNPh and FePzNPh was performed, we did not include a qualitative assessment into the manuscript.

For the study, we used solutions of photosensitizers that were stored for no more than three months, the stability of the compounds was confirmed by the absence of sediment and the preservation of their spectral properties and photodynamic activity.

Point 7: Study missing to prove that these compounds release ROS, which is essential for PDT.

Response 7: In the present work we aim to analyze the photophysical and photobiological properties of PzNPh and FePzNPh. The study of reactive oxygen species (ROS) generation was not included in the objectives of our study, which is primarily associated with the peculiarities of the photophysical properties of porphyrazines. As mentioned in the article, porphyrazines belong to the class of fluorescent molecular rotors. The probability of formation of particular states of the pz molecule (excited singlet planar and excited singlet twisted, with one to four twisted groups, triplet planar and twisted) are strongly dependent on viscosity, as well as their lifetimes. The common methods based on measuring in low viscous organic or water solutions lead to very significant underestimation of the singlet oxygen quantum yield, up to inability to detect them. Thus, to provide relevant information, a large-scale investigation in the physical chemistry of the processes and their dependence on viscosity must be performed.

Author Response

Dear Reviewer,

We would like to express our sincere appreciation for your careful attention to our manuscript and for the suggested improvements and valuable comments. We have revised the manuscript according to your remarks. Please find below the detailed description of the revision with our point-by-point answers.

Reviewer: The paper describes an investigation of cyanoarylporphyrazine and its iron complex as a photosensitizer (PS). The PS and iron complex were then studied for cellular toxicity, uptake and fluorescence properties. The paper is well-written and very easy to follow. The results are clearly presented, and conclusions are supported by the results. Characterization and analysis appear to be well-organized and systematic too.

Point 1: The major concern is about the originality/novelty of this work. The synthesis of iron complex of cyanoarylporphyrazine is neither very difficult task nor it very interesting from synthetic aspect.

Response 1: In the present work, the synthesis of the iron complex of tetra(2-naphthyl)tetracyanoporphyrazine (FePzNPh) is described for the first time and a comparative analysis of the photophysical and photobiological properties of the porphyrazine free base and its iron complex is carried out, with emphasis on the effect of iron on the studied properties. In particular, the spectral properties were studied, the quantum yield of fluorescence and photobleaching was determined, the dynamics of accumulation and photodynamic activity in relation to tumor cells in culture were studied.
We believe that the ease of obtaining compounds, including FePzNPh studied in this work (single-stage synthesis, mild conditions), is an additional advantage, as it provides reproducibility, easy production scalability, low product cost, and reliable quality control. We believe that interest in the synthesis of such compounds lies in the field of prospects for its practical implementation for production.

Point 2: Additionally, it is expected that iron complexation enhances the photodynamic therapy for the generation of toxic hydroxyl radicals by Fenton reaction (as described by the authors in the discussion). However, this was not observed by the authors and inadequate explanation was provided.

I believe this work can be accepted after addressing the concerns about novelty and significance of iron complexation in photodynamic activity.

Response 2: The presence of iron at the center of the photosensitizer macrocycle strongly affects its photochemical, photophysical, and photobiological properties [1], such as water solubility, triplet state lifetime, intersystem crossing efficiency, quantum yield and fluorescence lifetime, and photostability [2–3]. Because of this, the photobiological properties also change. It is these considerations that motivate our work.
In addition, in the presence of iron in the macrocycle, the development of the Fenton reaction can be expected. We have covered this aspect in the discussion (Lines 415-420 in the revised version of the manuscript). Our interest was also aroused by recent publications on the role of iron-catalyzed enhancement of the antitumor effect due to the induction of ferroptosis as an immunogenic cell death, which can effectively suppress tumor resistance to therapy [4]. However, this mechanism requires a separate study and is beyond the scope of the article.

References:

  1. Wu, Y.; Li, S.; Chen, Y.; He, W.; Guo, Z. Recent advances in noble metal complex based photodynamic therapy. 485 Chemical Science 2022, 13, 5085-5106, doi:10.1039/d1sc05478c
  2. Ali, H.; van Lier, J.E. Metal complexes as photo- and radiosensitizers. Chem Rev 1999, 99, 2379-2450
  3. Josefsen, L.B.; Boyle, R.W. Photodynamic Therapy and the Development of Metal-Based Photosensitisers. 488 Metal-Based Drugs 2008, 2008, 276109, doi:10.1155/2008/276109
  4. Mishchenko, T.A.; Balalaeva, I.V.; Vedunova, M.V.; Krysko, D.V. Ferroptosis and Photodynamic Therapy 580 Synergism: Enhancing Anticancer Treatment. Trends Cancer 2021, 7, 484-487

Reviewer 4 Report

Dear authors, you can find my disagreements, questions and remarks in the PDF attached (in order to streamline the process, I used the label/comment built-in tool throughout the text instead of listing the manuscript lines).

Some issues are fairly trivial and can easily be addressed, e.g. formatting and English language adjustments all through the manuscript, changing the abstract and improving/correcting the introductory section, amending the bibliographic references, removing the supporting material, and completing some more trivial information that is lacking. Apart from this, my main concerns are as follows.

Since the free-base porphyrazine PzNf was already synthesized and analyzed by you in ref. 33, which makes iron complex FePzNf the only novel chemical entity in this study, saying repeatedly that you present two new cyanoarylporphyrazines is simply untrue. Moreover, even if the synthetic strategy was not new and just an adaptation (similar to the Gd complex in ref. 44), a very brief description of the synthesis should be given. More significantly, FePzNf should be the subject of full structural characterization (IR, MS and elemental analysis or high resolution MS), not just the UV-Vis absorption and fluorescence emission details. I believe this was carried out, but the information is lacking.

Regarding the photodynamic activity, your previous work in ref. 33 specifies different values for PzNf, against the same cell type, and under equal experimental conditions (IC50 = 0.14 uM, IC50_dark = 9.5 uM, PDI = 68). Hence, not only did you previously synthesized PzNf, but its antitumor activity against A431 cells was also already presented, among other properties. It makes no sense to me, unless the previous data cannot be trusted and these new values are to be considered. In that circumstance, a statement must be made, and a correction issued to that journal. If not, then the previously published information must be showed in this paper, along with a clear mention to ref. 33.

FePzNf only presents a 3.5-fold decrease in dark toxicity and an increase in the photodynamic index - PDI (“which almost doubles compared to the free base and reaches a value of 60”), if we consider the new phototoxicity and photodynamic activity values of PzNf as true, not the previously published ones that I mentioned above. This means that one of your considerations for the promising potential of FePzNf might not exist.

I also would have liked you to select and test different tumor cells. This would be a step forward, concerning your previous and extended body of work on these matters, in which A431 cells were the core cell type employed by far and large.

Author Response

Dear Reviewer,

We would like to express our sincere appreciation for your careful attention to our manuscript and for the suggested improvements and valuable comments. We have revised the manuscript according to your remarks. Please find below the detailed description of the revision with our point-by-point answers.

Reviewer: Dear authors, you can find my disagreements, questions and remarks in the PDF attached (in order to streamline the process, I used the label/comment built-in tool throughout the text instead of listing the manuscript lines).

Point 1: Some issues are fairly trivial and can easily be addressed, e.g. formatting and English language adjustments all through the manuscript, changing the abstract and improving/correcting the introductory section, amending the bibliographic references, removing the supporting material, and completing some more trivial information that is lacking.

Response 1:

We have made changes to the Title, Abstract, Introduction, Materials and Methods, Results, and Discussion of the article. We fixed the spelling of superscripts and subscripts.
We have updated the bibliographic list by removing duplicate references. We have removed the supporting material by transferring the information to the main text of the article. Using the advice of a reviewer, we edited the English language. In addition, we rely on editing by MDPI's translators.

Point 2: Since the free-base porphyrazine PzNf was already synthesized and analyzed by you in ref. 33, which makes iron complex FePzNf the only novel chemical entity in this study, saying repeatedly that you present two new cyanoarylporphyrazines is simply untrue. Moreover, even if the synthetic strategy was not new and just an adaptation (similar to the Gd complex in ref. 44), a very brief description of the synthesis should be given.

Response 2: We have added a detailed description of the synthesis of PzNPh and FePzNPh  (Lines 129-139 in the revised version of the manuscript).

The preparation  of gadolinium (III) сomplex  previously published in ref.[1] differs from that used for Fe (II) complex. Nevertheless, the both approaches are united by the ease of cation chelation. The presence of strongly electron-deficient СN groups in the macrocycle framing provides relatively high acidity of the both NH groups compared the other tetrapyrole macrocycles.  It means that in polar solvent  (like water) cyanoarylporphyrazines exist mainly as free dianions which enable quick chelation  of any metal cations at room temperature.

References: 1.  Yuzhakova D.V., Lermontova S.A., Grigoryev I.S., Muravieva M.S., Gavrina A.I., Shirmanova M.V., Balalaeva I.V., Klapshina L.G., Zagaynova E.V. In vivo multimodal tumor imaging and photodynamic therapy with novel theranostic agents based on the porphyrazine framework-chelated gadolinium (III) cation. Biochimica et Biophysica Acta (BBA) - General Subjects. 1861(12), 3120-3130 (2017).

Point 3: More significantly, FePzNf should be the subject of full structural characterization (IR, MS and elemental analysis or high resolution MS), not just the UV-Vis absorption and fluorescence emission details. I believe this was carried out, but the information is lacking.

Response 3: We have added IR spectra and EI MS data  for FePzNPh to the article. (Lines 141-150 in the revised version of the manuscript).

Analytical data for PzNPh has been presented previously [1]. Briefly, IR spectrum, ν, cm–1: 3443 (N–H), 2200 (С≡N), 1595, 1508, 1496, 1469, 1451, 1435 (C=C, C=N), 902, 864, 839, 814, 749, 617 (C–HAr). Mass spectrum, m/z 918 [М]+. Found, %: C78.78; H 3.19; N 18.03. C60H30N12. Calculated, %: C 78.42; H 3.29; N 18.29. Yield 42%

References: 1. Lermontova, S.; Grigoryev, I.; Peskova, N.; Ladilina, E.; Lyubova, T.; Plekhanov, V.; Grishin, I.; Balalaeva, I.; Klapshina, L. Cyano-Aryl Porphyrazine Pigments with Polycyclic Substituents as the Promising Agents for Photodynamic Therapy and Potential Sensors of Local Viscosity. Macroheterocycles 2019, 12, 268-275, doi:10.6060/mhc190865k

Quotation:

IR spectra in mineral oil suspensions were recorded using an FSM 1201 spectrometer.

IR (KBr, νmax/cм-1): 2197 (С≡N); 1625 (shoulder), 1595 (shoulder), 1571, 1557, 1541, 1414 (C=N; C=C); 1300, 1261, 1215 (Car –H, C=N); 1046, 1027, 970, 952 (Car –H; С=С).

Positive ion electron ionization mass spectra were measured on a PolarisQ/TraceGCUltra GC/MS spectrometer at 70 eV in the mass number range of 50-1000.

EI MS (70eV): m/z (%) 369 [M+H2O-2Nf]2+  (35), 456 [M+H2O-3CN]2+ (100), 482 [M+H2O-CN]2+ (15), 504 [M+2H2O]2+ (22).

Point 4: Regarding the photodynamic activity, your previous work in ref. 33 specifies different values for PzNf, against the same cell type, and under equal experimental conditions (IC50 = 0.14 uM, IC50_dark = 9.5 uM, PDI = 68). Hence, not only did you previously synthesized PzNf, but its antitumor activity against A431 cells was also already presented, among other properties. It makes no sense to me, unless the previous data cannot be trusted and these new values are to be considered. In that circumstance, a statement must be made, and a correction issued to that journal. If not, then the previously published information must be showed in this paper, along with a clear mention to ref. 33.

Response 4: In our previously published article [1], the IC50 value for tetra(2-naphthyl)tetracyanoporphyrazine when incubated in the dark was 9.5 μM confidence interval [7.70-18], in the light 0.14 μM confidence interval [0.11-0.18]. As part of this study, a new batch of dye was synthesized. To validate the properties of the compound, we conducted an independent toxicity experiment, which resulted in IC50 values for the free base, which are consistent with previously obtained data, taking into account the statistical variation of the data: dark toxicity of 7.04 μM [6,13-8,16] and light toxicity 0.19 μM [0.16-0.21]. The results obtained confirmed the reproducibility of the synthesis. There are no statistical differences and contradictions between the data of the two articles.

References: 1. Lermontova, S.; Grigoryev, I.; Peskova, N.; Ladilina, E.; Lyubova, T.; Plekhanov, V.; Grishin, I.; Balalaeva, I.; Klapshina, L. Cyano-Aryl Porphyrazine Pigments with Polycyclic Substituents as the Promising Agents for Photodynamic Therapy and Potential Sensors of Local Viscosity. Macroheterocycles 2019, 12, 268-275, doi:10.6060/mhc190865k

Point 5: FePzNf only presents a 3.5-fold decrease in dark toxicity and an increase in the photodynamic index - PDI (“which almost doubles compared to the free base and reaches a value of 60”), if we consider the new phototoxicity and photodynamic activity values of PzNf as true, not the previously published ones that I mentioned above. This means that one of your considerations for the promising potential of FePzNf might not exist.

Response 5: We made additional experiments to study the photodynamic activity of PzNPh and FePzNPh on cell lines HaCaT and SKOV-3.ip. For all three cell lines, the PDI was greater for the iron complex than for the free base. Thus, we consider our assertion to be valid. Differences in the values of the PDI  in different experiments are due to statistical variations when working with biological objects.

Point 6: I also would have liked you to select and test different tumor cells. This would be a step forward, concerning your previous and extended body of work on these matters, in which A431 cells were the core cell type employed by far and large.

Response 6: We did additional experiments on the comparative analysis of the toxicity of PzNPh and FePzNPh on HaCaT and SKOV-3.ip cell lines. The results obtained are included in the manuscript. (Lines 369-379 in the revised version of the manuscript).

Point 7: "COMPARATIVE ANALYSIS OF NEW CYANOARYLPORPHYRAZINE AND ITS IRON COMPLEX AS PHOTOSENSITIZERS FOR ANTICANCER PHOTODYNAMIC THERAPY" this is misleading, since the free-base porphyrazine was already prepared and assessed by your group. the iron complex and its properties is, in fact, novelty.

Response 7: Corrected 

Quotation: COMPARATIVE ANALYSIS OF TETRA(2-NAPHTHYL)TETRACYANOPORPHYRAZINE AND ITS IRON COMPLEX AS PHOTOSENSITIZERS FOR ANTICANCER PHOTODYNAMIC THERAPY

Point 8: "A leading trend today is the search for new effective photodynamic agents." I am not sure this is a'leading trend'. A lot of well known 'old' and already approved PSs are also being studied in recent times, for instance in novel nanoformulations, biological carriers, polymer supports, etc. light/laser systems used on PDT, regardless of the PS (novel or known), are also being studied.

Response 8: Corrected. (Lines 13-15 in the revised version of the manuscript). 

Quotation: The main trends today are the search for new effective photodynamic agents and creation of targeted delivery systems with the function of controlling the release of the agent in the tumor. 

Point 9: "Previously, our research team synthesized and studied compounds from the cyanoporphyrazine group that are both photosensitizers and sensors of the local microenvironment. With their help, we demonstrated the possibility of assessing the release of photosensitizer from the transport carrier in real time in  vivo" addressing your previous work on the subject at hand, although important for a more global insight into your research efforts, is something to be done in the introduction and discussion sections, not the abstract.

Response 9: Corrected. (Lines 12-27 in the revised version of the manuscript). 

Quotation: Photodynamic therapy (PDT) is a rapidly developing modality of primary and adjuvant anticancer treatment. The main trends today are the search for new effective photodynamic agents and creation of targeted delivery systems with the function of controlling the release of the agent in the tumor. Recently, the new group of cyanoarylporphyrazine dyes was reported which combine the properties of photosensitizers and sensors of the local microenvironment. Such unique characteristics allow assessing the release of photosensitizer from the transport carrier in real time in vivo. The aim of the present work was to compare the photophysical and photobiological properties of tetra(2-naphthyl)tetracyanoporphyrazine and its newly synthesized Fe(II) complex. We have shown that the chelation of Fe(II) cation with the porphyrazine macrocycle leads to a decrease in molar extinction and an increase in quantum yield of fluorescence and photostability. We demonstrate that the iron cation significantly affects the rate of the dye accumulation in cells, the dark toxicity and photodynamic activity, and the direction of the changes depends on the particular cell line. However, in all the cases, the photodynamic index of a metal complex was higher than that of a metal-free base. In general, the both the compounds are found to be very promising for PDT, including the use with transport delivery systems, and can be recommended for further in vivo studies.

Point 10: "Current research has shown that iron notably impacts the photophysical properties of cyanoporphyrazine: in the presence of iron, the quantum yield of fluorescence and the photostability of porphyrazine increases. It has also shown that the iron cation significantly affects the photobiological characteristics. For instance, the photodynamic index of a metal complex is higher than that of a metal-free base, but the rate of its accumulation in cells is lower." again, this is introduction material. you should synthesize your paper's objectives, goals, methods and main results in the abstract section!

Response 10: Corrected.  (Lines 12-27 in the revised version of the manuscript).

Point 11: "Photodynamic therapy was developed over forty years ago and is now widely used in treating various types of cancer" this is untrue! it is widely used to treat some skin conditions and more superficial or easily accessible tumors in the clinic, along with pre- or post-surgical resection/chemotherapy treatments.

Response 11: Photodynamic therapy was developed over forty years ago and is now widely used in treating various types of cancer. 
Among the most important advantages of PDT is low systemic toxicity because the cytotoxic effect of the PS in the absence of light irradiation is several orders of magnitude lower than in the irradiated cancer tissue. But a current limitation is the mode of light delivery, which is currently restricts treatment to superficial tumors or tumors accessible with an endoscope: non-melanoma skin cancer, retina and ocular malignancies, cancer of the digestive tract, urinary system malignancies, tracheobronchial malignancies, and cervical cancer. Also, PDT was shown to be effective in improving patient prognosis when used after surgical resection of glioblastoma. Large solid tumors and tumors located deep in brain and parenchymal organs require interstitial PDT (IPDT). This approach involves delivery of light by optic fibers inserted via needles or catheters. IPDT has proven to be successful in the treatment of primary and recurrent prostate cancer, breast cancer, glioma, head and neck tumors, and pancreatic cancer. The progress in light delivery systems implies expansion of the list of tumor localizations that can become accessible to PDT. An elegant technical solution is the use of wireless optoelectronic devices fixed in the body to provide local continuous low-power irradiation of a tissue for up to several days, an approach named metronomic PDT [1].

Reference: 1. Mishchenko, T., Balalaeva, I., Gorokhova, A. et al. Which cell death modality wins the contest for photodynamic therapy of cancer?. Cell Death Dis 13, 455 (2022). https://doi.org/10.1038/s41419-022-04851-4].

Point 12: "This therapeutic procedure involves administration of a photosensitive compound (a photosensitizer, a PS), which target cells After absorption, tumor cells are exposed to the visible light of the wavelength that corresponds to the peaks of the PS excitation spectrum" I would refrain the use of the term 'absorption' and 'absorbs'. the mechanisms of internalization might differ according to PS, tumor type, loading vehicle, etc...

Response 12: Corrected. (Lines 37-38 in the revised version of the manuscript). 

Quotation: This therapeutic procedure involves administration of a photosensitive compound (photosensitizer, PS), which accumulates in tumor. Then, tumor cells are exposed to visible or near infrared (NIR) light of the wavelength that corresponds to the peaks of the PS absorption spectrum. 

Point 13: "In vitro and in vivo studies demonstrate that physico-chemical properties of PS affect significantly the processes of Ps localization and its photodynamic activity in cells and tumor tissue [6]". italic formatting, here and throughout the text.

Response 13: Corrected. (Line 52 in the revised version of the manuscript).

Point 14: "They are TOOKADS® soluble (Padeliporfin, WST11), which is a palladium-coordinated bacteriochlorophyll a, approved for the treatment of prostate cancer in Mexico, Israel and over 30 EU countries [25] and Photosens®, which is sulfonated aluminum phthalocyanine approved for the treatment of skin, liver, breast, lung and gastrointestinal cancer in Russia [26]" TOOKAD

Response 14: Corrected. (Line 69 in the revised version of the manuscript).

Point 15: "They have a number of advantages connected with maintenance of solubility, increase in time of circulation of PS in blood, an opportunity of the address delivery and controlled release of PS in an area of interest and a number of other advantages [32]". please rephrase, this is confusing.

Response 15: Corrected. (Lines 80-93 in the revised version of the manuscript). 

Quotation: Another approach to increasing the effectiveness of PDT is the development of delivery systems for photosensitizers [32]. Nanoscale carriers are of particular interest as delivery systems. They can provide an enhancement in PS solubility and prolonged blood circulation of PS, facilitate PS targeted delivery and controlled release in the pathological site [33-37]. Targeted accumulation of photosensitizers in the tumor helps to reduce the serious side effects of PDT, such as prolonged (over several months) photosensitivity of the skin and eyes . Polymer particles and micelles [38], liposomes [39], gold, calcium and silica nanoparticles [40-42] are especially popular among the nanoparticles used for the delivery of PSs. Additionally, nanoparticles can be modified with stimulus-sensitive moieties. Stimulus-sensitive carriers are able to respond to specific factors of tumor microenvironment which leads to the disintegration of the carrier in the tumor site and controlled release of the drug. The influencing factors can be of exogenous (magnetic field, heating, exposure to UV and IR radiation or ultrasound) [32] and endogenous nature – changes in the pH level [41], enzymes and redox potential [43].

Point 16: "In the works [40] and [41] demonstrated for the first time that cyanoporphyrazines are successfully loaded into amphiphilic polymer brushes and, due to the presence of such a container, selectively accumulate and retain in tumors, providing subsequent successful photodynamic therapy." please rephrase like exemplified in the previous line. this is not a propper way of presenting and referencing your previous studies.

Response 16: Corrected. (Lines 101-105 in the revised version of the manuscript). 

Quotation: The dyes of cyanoporphyrazine group were also successfully loaded into amphiphilic polymer brushes and, due to the presence of such a container, were selectively accumulated and retained in tumors, providing subsequent successful photodynamic therapy [51],[49]. The potential of cyanoporphyrazines was fully shown on the example of vaterite particles [52].

Point 17: "In this study, we perform a detailed analysis of two novel photosensitizers namely tetra(2-naphthyl)tetracyanoporphyrazine (hereinafter, PzNf) and its iron complex (hereinafter, FePzNf) in order to assess their possible application in PDT and evaluate the effect of adding an iron metal atom in the center of the porphyrazine macrocycle." again, the novelty is in the iron complex, not the free-base PS. please rephrase.

Response 17: Corrected. (Lines 111-114 in the revised version of the manuscript). 

Quotation: In this study, we perform a detailed analysis of two photosensitizers, namely, tetra(2-naphthyl)tetracyanoporphyrazine (hereinafter, PzNPh) and its iron complex (hereinafter, FePzNPh) in order to assess their possible application in PDT and evaluate the effect of adding an iron metal atom in the center of the porphyrazine macrocycle. 

Point 18: Materials and Methods. a statistical analysis subsection should be provided in this section.

Response 18: Corrected. (Lines 243-245 in the revised version of the manuscript). 

Quotation: 

2.8. Statistical analysis

Statistical analysis was performed in GraphPad Prism (v.9.0) (GraphPad Software, USA). Cell death was analyzed by ANOVA followed by Dunnett’s test.

Point 19: "The synthesis of the former had been described  previously [33,43] and the latter had been prepared by ferrous acetate mixing with free-base in inert atmosphere that is similarly to gadolinium cyanoarylporphyrazine complex that had been reported by us earlier [44]." I understand that a previously used method was chosen to prepare this iron complex, as referenced, but since the result is a novel previously unpublished molecule, than a very brief description of the synthesis should be given and, more importantly, FePzNf should be fully and structurally characterized (IR, MS and elemental analysis or high resolution MS), not just the UV-Vis and fluorescence data.

Response 19: Corrected. (Lines 129-139; 141-150 in the revised version of the manuscript). 

Point 20: "Absorption was recorded in the range of 300–700 nm, fluorescence was recorded in the range of 600–800 nm with excitation at a wavelength of 580 nm." or is it 570 nm, as stated in Figure 2?

Response 20: Corrected. (Line 283 in the revised version of the manuscript). 

Quotation: Absorption (a) and fluorescence (b) spectra of PzNPh and FePzNPh (both at 5 µM) in glycerol. The fluorescence was excited at λex 570 nm.

Point 21: "The analysis of photophysical properties of PSs has revealed that both porphyrazines absorb and fluoresce in the optical window in biological tissue 500-650 nm and 600-800 nm, respectively (Figure 2, Figure S1, Table 1)." figure S1 could be added to figure 2, canceling the supporting information file.

Response 21: Corrected. (Lines 258-261 in the revised version of the manuscript). 

Point 22: any account on why these values differ from the ones previously presented by you in [33]?! also, this is not a novelty feature of the paper.

Response 22: This issue has been explained above.

Point 23: "To estimate the photobleaching coefficient of photosensitizers, we used a solution of PzNf and FePzNf in glycerin. Glycerol was used as a solvent in the assessment of photobleaching because of the photophysical features of the studied dyes. When energy is transferred to them in the form of a quantum of light, the resulting energy of the excited state  can be spent through three main channels - intramolecular rotation, fluorescence, and intersystem conversion." I believe you mean internal conversion, the radiationless deactivation mode. intersystem crossing occurs with e.g. molecular oxygen, leading to the active antitumoral ROS.

Response 23: Corrected. (Lines 293-296 in the revised version of the manuscript). 

Quotation: When energy is transferred to them in the form of a quantum of light, the resulting energy of the excited state can be spent through three main channels - intramolecular rotation, fluorescence, and intramolecular charge transfer.

Point 24: "Figure 3. Photobleaching of PzNf and FePzNf in the glycerol solution under irradiation, a – photobleaching of PzNf; b – photobleaching of FePzNf; с – photobleaching of PzNf and FePzNf at maximum" please rephrase as this looks a bit incomplete. Also, the statistical data is missing "The data are presented as the mean values ± SD (n = ...)".

Response 24: Corrected. (Lines 308-311 in the revised version of the manuscript). 

Quotation: Photobleaching of PzNPh and FePzNPh in the glycerol solution under irradiation, a – Absorption spectra of PzNPh after irradiation; b – Absorption spectra of FePzNPh after irradiation; с – The dose-dependent decrease in optical density at the absorption maximum for PzNPh and FePzNPh. The data are presented as the mean values ± SD (n = 3)

Point 25: "The choice of the cell line was determined by the fact that the PDT method is used to treat skin cancer [50,51]." this is a weak argument. PDT is being studied in all sorts of cell types, and even clinically aplied to a few other non-epidermal conditions and tumors.

Response 25: Corrected. (Lines 315-316 in the revised version of the manuscript). 

Quotation: The choice of the cell line was due to the fact that the main area of PDT application is superficial tumors, including skin cancer.[59,60].

Point 26: figures 4a and 4b lack the length units in the scale bar.

Response 26: Corrected. (Line 329 in the revised version of the manuscript). 

Quotation: Analysis of the cellular uptake of PzNPh and FePzNPh. Representative confocal images of human epidermoid carcinoma A431 cells after 15 min (a) and 240 min (b) of incubation with PzNPh and FePzNPh (5 µM). Relative accumulation of the PzNPh and FePzNPh after 15 and 240 min incubation estimated from a confocal microscopy experiment (c) (n ≥ 10). The values are given minus the background, which did not exceed 4 a.u. Scale bar 20 µm. The data are presented as the mean values ± SD 

Point 27: "When adding iron, the photodynamic index almost doubled and reached a value of 60." this is no longer the case if we take into account the figures your own group previously published in [33] regarding PzNf, that is IC50 = 0.14 uM, IC50_dark = 9.5 uM, PDI = 68!!, using exactly the same experimental setup conditions (4h incubation, 615…635 nm, 20 mW/cm2, 20 J/cm2)!!

Response 27: We agree that the precise values of an increase in PDI strongly depend on variability on biological data between independent experiments (and reproducibility of the synthesis). We deleted the term “doubled” to avoid misleading. (Lines 362 in the revised version of the manuscript). 

Quotation: For FePzNPh, it was significantly higher and reached a value of 60.

Point 28: again (similarly to table 1), your previous work in [33] states different values against the same cell type and under equal irradiation conditions, i.e. IC50 = 0.14 uM, IC50_dark = 9.5 uM, PDI = 68 !! PzNf was already synthesized and studied by you and the data presented elsewhere... This is not new, although the data is different, and it makes no sense to me, unless the previous data can not be trusted. in that case a statement must be made, and also a correction issued to the journal. if not, then the previously published information must be indicated, along with a clear reference to [33]!!

Response 28: This issue has been explained above.

Round 2

Reviewer 2 Report

The authors have responded satisfactorily to some queries, but it is still confusing for others. 

The MASS and IR spectra are still missing, although their interpretations are included. I'm not sure why proton and carbon NMR were used to realize the synthesis and purification. 

Figure 2 legend needs corrections. 

For response 3: Glycerin-ethanol mixture was not used in the study, then how is this response justifiable? 

For response 7: If ROS is not relevant to the study, then the author must change the title and revise the introduction section by skipping PDT.

Author Response

Dear Reviewer,

We would like to express our sincere appreciation for your careful attention to our manuscript and for the suggested improvements and valuable comments. We have revised the manuscript according to your repeated remarks. Please find below the detailed description of the revision with our point-by-point answers.

Reviewer: The authors have responded satisfactorily to some queries, but it is still confusing for others.

Point 1: The MASS and IR spectra are still missing, although their interpretations are included. I'm not sure why proton and carbon NMR were used to realize the synthesis and purification.

Response 1: We apologize for not having previously demonstrated the MASS and IR spectra of FePzNPh. The spectra that we describe in the article are presented (Figure 1S and 2S).

 We present data for proton NMR below:

1H NMR (DMSO-d6) of FePzNPh

δH ppm: 7,57 (br. s, 3H, -NPh), 8,00 (br. m, 4H, -NPh).

Unfortunately, we unable to present 13C NMR for FePzNPh since  its solubility in DMSO-d6 is not enough to provide FePzNPh  concentration which is necessary for good quality  spectrum.

Point 2: Figure 2 legend needs corrections.

Response 2: Corrected. (Lines 291-292 in the revised version of the manuscript).

Quotation: Figure 2. Absorption and fluorescence spectra of PzNPh and FePzNPh in glycerol (a, b) and in water (c, d) (both at 5 µM). The fluorescence was excited at λex 570 nm.

Point 3: For response 3: Glycerin-ethanol mixture was not used in the study, then how is this response justifiable?

Response 3: We apologize for confusing the Reviewer with incorrect answer. During the paper preparation, we performed a series of experiments on ethanol-glycerol mixtures, but the data were excluded from the final version of the manuscript.

The correct explanation will be the following.

 It was reported that viscosity influences the photo-activity of dyes. Thus, how are the results in glycerol relevant to clinical testing?

The studied compounds (PzNPh and FePzNPh) are characterized by the dependence of the lifetime of the excited state and the quantum yield of fluorescence on the viscosity of the medium. We chose glycerol as the viscous medium. Viscous media are closer in their physical properties to real conditions in a cell than aqueous solutions. For example, the viscosity of mesenchymal stem cell membranes during their differentiation can reach 860 cP, which is comparable to the viscosity of glycerol [Kashirina, A.S., López-Duarte, I., Kubánková, M. et al. Monitoring membrane viscosity in differentiating stem cells using BODIPY-based molecular rotors and FLIM. Sci Rep 10, 14063 (2020). https://doi.org/10.1038/s41598-020-70972-5]

Point 4: For response 7: If ROS is not relevant to the study, then the author must change the title and revise the introduction section by skipping PDT.

Response 4: T.J. Doherty defines a photosensitizer as an agent that is activated by light of a certain wavelength, which leads to a sequence of photochemical and photobiological processes that cause irreversible photodamage to tumor tissues. [Dougherty T.J., Homer C.J., Henderson B.W., Jory G., Kessel D., Korbelik M., Moan J., Peng K. Photodynamic therapy. J Natl Cancer Inst. 1998 June 17; 90(12): 889-905. doi: 10.1093/jnci/90.12.889. PMID: 9637138; PMCID: PMC4592754].

Thus, the basis for attributing the compound to PS group is its biological activity. This point of view is supported by the PDT/PS definition provided by the world-known leading scientists in this field. We, therefore, can name the compounds studied as PSs, since their photodynamic activity is experimentally proven against several cell types. In the presented manuscript, we have demonstrated that PzNPh and FePzNPh photobleach when irradiated with light. In addition, we have shown that PzNPh and FePzNPh are characterized by high light activity and low dark toxicity against three different tumor cell lines. In our opinion, this fact convincingly proves that PzNPh and FePzNPh are photosensitizers.

Nevertheless, we performed additional experiments to prove singlet oxygen production by PzNPh and FePzNPh under light irradiation (Lines 243-248, 371-373 and Figure S3)

Quotation:
We also conducted a study on the production of reactive oxygen species by a chemical trap method using 1,3-diphenylisobenzofuran (DPBP) at a concentration of 200 μM in an ethanol-glycerol mixture (50%). Combinations of PzNPh and FePzNPh 5 μM were used for measurements. The solutions were irradiated with light at a dose of 0 to 20 J/cm2 using an LED light source (λex 615–635 nm, 20 mW/cm2). The absorbance of solutions at a wavelength of 420 nm was measured using a Synergy MX plate reader. (Lines 243-248)

A study of the production of reactive oxygen species (Figгку 3S) showed a higher generation of reactive oxygen for PzNPh, which is consistent with the data presented above. (371-373)

Reviewer 4 Report

Dear authors, thank you for the revised document. I acknowledge that the manuscript was fairly improved and that important and missing information was added, namely synthetic details and structural characterization of the new compound FePzNf). I was also pleased for the addition of two more cell types to your workflow regarding the (photo)cytotoxicity assessment. I hope that you will continue to do so in your following studies (either these or other tumor and normal cells, although I believe the SKOV-3.ip cells deserve further future evaluation). In fact, both the photodynamic activity and the pdi figures were better in the new ovarian adenocarcinoma cell line for the iron complex, comparing to the free base, but also regarding the ‘old’ A431 cells.

However, my problem with the ‘new’ photodynamic activity and phototoxicity figures of previously reported PzNf was not so much the statistical differences and possible contradictions (I agree with your rebuttal that the IC50 and CI95 values are all in agreement with the previous ones and, hence, demonstrated reproducibility). The main issue is the lack of novelty in that regard and the repetition of previous work done recently by you in ref.44 (ref.33 in version 1 of manuscript, i.e. Macroheterocycles 2019 12(3) 268-275 / doi: 10.6060/mhc190865k).

Together with the synthesis of the PzNf photosensitizer, also the photophysical properties in water (not in glycerol, that is in fact new), the dark/photoinduced cytotoxicity, the accumulation profile via fluorescence intensity, and the cellular uptake (measured by confocal imaging) evaluations in A431 cells was all already done by you in ref.44. You chose to repeat them in this manuscript, with no mention to that fact whatsoever. All of the studies mentioned above are not new research, but provide new data in some circumstances, and to reply that a new batch of photosensitizer PzNf was prepared, so it was decided to conduct many of the same assays using the same experimental setup, without acknowledging it in the manuscript can be viewed, at the very least, as misleading the reader.

A couple more comments and suggestions were made regarding the results obtained with the novel cell lines employed in version 2 of the manuscript. They can be found in the revised pdf file attached (much in the same manner as with the first draft).

Author Response

Dear Reviewer,

We would like to express our sincere appreciation for your careful attention to our manuscript and for the suggested improvements and valuable comments. We have revised the manuscript according to your repeated remarks. Please find below the detailed description of the revision with our point-by-point answers.

Reviewer: Dear authors, thank you for the revised document. I acknowledge that the manuscript was fairly improved and that important and missing information was added, namely synthetic details and structural characterization of the new compound FePzNf). I was also pleased for the addition of two more cell types to your workflow regarding the (photo)cytotoxicity assessment. I hope that you will continue to do so in your following studies (either these or other tumor and normal cells, although I believe the SKOV-3.ip cells deserve further future evaluation). In fact, both the photodynamic activity and the pdi figures were better in the new ovarian adenocarcinoma cell line for the iron complex, comparing to the free base, but also regarding the ‘old’ A431 cells.

Point 1 However, my problem with the ‘new’ photodynamic activity and phototoxicity figures of previously reported PzNf was not so much the statistical differences and possible contradictions (I agree with your rebuttal that the IC50 and CI95 values are all in agreement with the previous ones and, hence, demonstrated reproducibility). The main issue is the lack of novelty in that regard and the repetition of previous work done recently by you in ref.44 (ref.33 in version 1 of manuscript, i.e. Macroheterocycles 2019 12(3) 268-275 / doi: 10.6060/mhc190865k). Together with the synthesis of the PzNf photosensitizer, also the photophysical properties in water (not in glycerol, that is in fact new), the dark/photoinduced cytotoxicity, the accumulation profile via fluorescence intensity, and the cellular uptake (measured by confocal imaging) evaluations in A431 cells was all already done by you in ref.44. You chose to repeat them in this manuscript, with no mention to that fact whatsoever. All of the studies mentioned above are not new research, but provide new data in some circumstances, and to reply that a new batch of photosensitizer PzNf was prepared, so it was decided to conduct many of the same assays using the same experimental setup, without acknowledging it in the manuscript can be viewed, at the very least, as misleading the reader.

Response 1:

To avoid misleading the reader, we added to the text the direct mention of the fact, that PzNPh was previously reported as well as explanation of the rationale for its inclusion to the text of the manuscript. Also, we added the comment on reproducibility the parameters between synthetic batches. The following links to the published data present in the revised manuscript:

  • Reference to the synthesis procedure (Line 132)
  • Reference to the analytical data (Line 151)
  • Reference to the spectral data (Table 2)
  • Reference to the data .about cellular uptake of PzNPh (Line 357)
  • Reference to cytotoxity data of PzNPh against A431 cells (Line 376)
  • New discussion on reproducibility of the characteristics (Lines 356-357, 376-378)

Quotations: 

The results are consistent with those obtained earlier for PzNPh [44]. (Lines 356-357)

Data on cytotoxicity for PzNPh did not differ from those obtained previously [44]. This demonstrates a high degree of reproducibility of the synthesis procedure and is important for further practical implementation. ( Lines 376-378)

Point 2:

A couple more comments and suggestions were made regarding the results obtained with the novel cell lines employed in version 2 of the manuscript. They can be found in the revised pdf file attached (much in the same manner as with the first draft).

Point 2.1. since you did not perform the cellular uptake assay for the new cells, there is no way of knowing 4 hours is the ideal incubation period for them, although I understand using the same time for comparison purposes.

Response 2.1: We agree, the Reviewer quite rightly noted that it would be more correct to evaluate the dynamics of PS uptake for each line separately. However, for comparison at this stage, we used a time point of 4 hours as in the case of A431 cell culture.

Point 2.2. The photodynamic activity was increased, not decreased, by the addition of the Fe metal center.

Response 2.2: Corrected (Lines 384-390)

Quotation: 

The cytotoxicity and photodynamic activity of both the compound were additionally analyzed against two cell lines of different origin: HaCat human keratinocytes and SKOV-3.ip human ovarian adenocarcinoma (Figure 6, Table 2). In fact, inclusion of an iron atom lead to varying effects depending of cell line. We explain this by the metabolic peculiarities of the selected cell lines due to their different origin. However, for both cell lines we observed an increase of photodynamic index, similarly to A431. For HaCat cells it increased by about 6-fold and for SKOV-3.ip by 3.7-fold compared to the freebase.

Point 2.3: It might be interesting and useful to use nM instead of uM units in the photocytotoxicity figures (190 vs 400 nM; 40 vs 34 nM; 180 vs 70 nM), given that these PSs are quite active (low IC50_light values). the fact that it is easier to compare and discuss in terms of nanomolar units is a proof of that exactly.

Response 2.3: We agreed and changed the dimension for light IC50 (Lines 367 and Table 2)

Point 2.4. Why decimal points just in this pdi value?

Response 2.4.: Corrected. We left an integer value of PDI in the Table 2

Point 2.5. This is only true for A431 cells, not for the SKOV-3.ip cell line, in which the iron complex  was better, i.e. more phototoxic (70 vs 180 nM).

Response 2.5. Corrected (Line 393)

Quotation: Despite the slight decrease in photoinduced toxicity for FePzNPh against A431 cells.

Point 2.6: it also improves the photodynamic activity against the ovarian adenocarcinoma cell line selected, comparing to PzNf (70 vs 180 nM), as opposed to the epidermoid carcinoma cells (190 vs 400 nM), although both exhibited low nanomolar values.

Response 2.6: The Reviewer correctly noted the increase in light toxicity. However, we did not observe this effect for each studies cell line. Therefore, we had focused on increase of the photodynamic index, the patterns for which are the same for all cell lines.

Quotation: Analysis of photodynamic activity has shown that addition of iron cation to macrocycle leads to the increase of photodynamic index (Table 2).

Round 3

Reviewer 2 Report

Authors addressed all the comments with satisfactory responses. The manuscript can be published.